# Evaluation of Dentinal Thickness and Remaining Dentine Volume around Root Canals Using Cone-Beam Computed Tomography Scanning

**DOI:** 10.3390/dj11050124

**Published:** 2023-05-04

**Authors:** Svetlana Razumova, Anzhela Brago, Haydar Barakat, Dimitriy Serebrov, Zoya Guryeva, Gleb S. Parshin, Vasiliy I. Troitskiy

**Affiliations:** 1Department of Propedeutics of Dental Diseases, Medical Institute, Peoples’ Friendship University of Russia (RUDN University), 6 Miklukho-Maklaya St., Moscow 117198, Russia; 2Department of Infectious Diseases, Institute of Clinical Medicine, I.M. Sechenov First Moscow State Medical University (Sechenov University), 2 Bolshaya Pirogovskaya St., Moscow 119435, Russia

**Keywords:** dentinal thickness, CBCT, root canal, remaining dentine volume

## Abstract

Background: The remaining dentinal thickness is a significant factor to deal with when planning post-endodontic treatment. Aim: To assess the changes in the root canal dentinal thickness of intact and endodontically treated teeth using CBCT scans in the coronal, middle, and apical third of the root canal. Material and methods: Three hundred CBCT scans for three age groups were analyzed to study the dentinal thickness pre- and post-endodontic treatment. The dentinal thickness (DT) was measured along the buccal, mesial, distal, and lingual/palatal walls from the inner surface of the root canal to the outer surface in mm. Statistical analysis was set at 0.05. Results: The results of this study showed that the buccal, palatal, distal, and mesial dentinal thickness in intact and endodontically treated teeth is different. The differences were statistically significant when comparing the parameters of “healthy” and “treated” teeth (*p* < 0.05). There were no statistically significant differences in indicators associated with age (*p* > 0.05). In the coronal third of the root canal, the least amount of dentin tissue lost was 4.2% for mandibular canines. Conclusions: The dentinal thickness in the coronal and middle third of the root decreases significantly more than the apical third. The most loss of dentine volume was among molar teeth, and the remaining dentinal thickness was less than 1 mm, which would increase the risk of complication while preparing the canal for a post.

## 1. Introduction

It is known that tooth structure is lost during root canal treatment, and teeth treated endodontically are more vulnerable to fracture than intact teeth [1,2]. To serve these teeth, it is critical to restore their shape and function after endodontic treatment. One of the most significant factors that affect post-endodontic treatment is the dentinal thickness that will remain after the canal preparation. With the diversity of root canal instrumentation systems, it is critical to take into consideration that dentinal thickness may differ after the root canal preparation. Clinicians should plan the endodontic treatment without the aggressive root canal preparation that leads to a decrease in the dentinal thickness, especially in cases of re-endodontic treatment. The complex anatomy of the root canal system and the presence of grooves and lateral canals in intact teeth, which have been reported to have less than 1 mm of dentinal thickness [3,4], are important issues to be considered while preparing the root canal, especially in the premolar and molar areas, to avoid thinning or perforating the dentine wall in these areas [5]. Dentinal thickness is also important when preparing the canal for a post, as several studies have suggested that the post should be surrounded by at least 1 mm of sound dentine [6,7,8].

To determine the changes in the amount of dentine removed during endodontic treatment, Bramante et al. [9] proposed to superimpose photographs of the root canal diameter before and after instrumentation and measure the deviations between the two contours. Quantification of post-operative deviation in the root canal diameter can be performed using the centering factor method [10,11,12] or by measuring dentine thickness before and after treatment [13]. This method can evaluate the circular removal of root dentine and the frequency of isthmuses [14].

The change in the canal volume after preparation is associated with instrumental and medical treatment [15,16,17]. After treatment, the increase in the volume of the internal lumen of the canal is proportionally higher in the coronal and middle third than in the apical one; this is because of the taper, the instruments used, and the application of force during preparation [18,19,20]. Clinically, an increase in root canal volume in the coronal and middle third allows for more efficient irrigation of the apical part, but at the same time suggests that apical mechanical debridement is not as effective as coronal debridement [18,21].

For studying the root canal system, modern diagnostic techniques include cone beam computed tomography (CBCT) and micro-CT [17,22,23]. The use of CBCT allows one to measure the volume of the root canal dentine and its surface area and to clarify the anatomy of the root canal before and after preparation. Micro-CT additionally makes it possible to assess the degree of the untreated canal surface in three dimensions.

The aim of this study was to assess the changes in the root canal dentinal thickness of intact and endodontically treated teeth according to CBCT scans in the coronal, middle, and apical third of the root canal.

## 2. Materials and Methods

This cross-sectional study was conducted to evaluate the dentinal thickness among different age groups in the period between November 2019 and March 2021.

Three hundred CBCT scans for 300 subjects aged 20–70 years were included in this study. The study was conducted in 5 dental clinics and radiologic diagnostic centers for three-dimensional radiological scanning in Moscow.

Patients who were diagnosed to have endodontic treatment needs (apical periodontitis, pulpitis, dental trauma) in these clinics and referred for CBCT were asked for their consent to participate in this study. After collecting the sample, CBCT scans were divided into three age groups with 100 subjects each: young adults (20–44) years, middle-aged adults (45–59) years, and the elderly (60–70) years.

The CBCT scans were taken using a 3D eXam (KaVo, Biberach, Germany) with standard exposure settings (23 cm × 17 cm field of view, 0.3 mm voxel size, 110 kv, 1.6–20 s) and were viewed by three examiners in a semi-dark room using I-CAT viewer software (version 10, Hatfield, UK).

After collecting CBCT, the scans were analyzed to determine the number of intact and endodontically treated teeth, regardless of the technique used to prepare the root canal by two endodontists. Intact and endodontically treated teeth were included in this study; teeth with resorption were excluded. Table 1 and Table 2 explain the number of teeth included in this study.

For calibration, a pilot study of 10 CBCT was performed. The analysis of dentinal thickness was performed independently by two endodontic researchers and repeated after a 2-week interval. A comparison of the DT results was performed between the observers, and in the case of disagreement, the case was discussed until a decision was reached. The Kappa value for the intra-observer agreement was 0.85 for both observers and 0.80 for the inter-observer agreement.

The measurement of dentinal thickness was performed on intact and endodontically treated maxillary and mandibular teeth using CBCT in the horizontal plane at the levels of the coronal, middle, and apical regions in all three age groups. The distance was measured in mm from the inner surface of the buccal, mesial, palatal/lingual, and distal sides of the root canal to the outer surface of the root (Figure 1 A–C). The loss of dentine tissue was evaluated in percent.

Statistical analysis: IBM SPSS Statistics v 22.0, a licensed package (IBM, Chicago, IL, USA), was used for the statistical processing of the study data. Descriptive statistics were used for the processing of the received data. For comparing the dentinal thickness among age groups and between intact and treated teeth, the independent student test was used. Statistical significance was set at 0.05.

## 3. Results

The results of this study showed that the buccal, palatal, distal, and mesial dentinal thicknesses in intact and endodontically treated teeth are different. The differences were statistically significant when comparing the parameters of “healthy” and “treated” teeth (*p* < 0.05). There were no statistically significant differences in indicators associated with age (*p* > 0.05).

### 3.1. Measurment of DT of Maxillary Teeth

The measurements of DT of the root canals of central incisors in mesial, distal, buccal, and palatal sides revealed that the volume of dentine decreased from 3.6% to 27.7%, and for lateral incisors, the volume of excised dentine was from 1.7% to 20.4%. For canines, the volume of dentinal tissues decreased from 3.6% to 17.3% after root canal preparation.

For maxillary first premolars, there was a significant difference in the DT in all age groups. The loss of dentine along mesial, distal, buccal, and palatal sides ranged from 3% to 31%. For maxillary second premolars, the measurements revealed that the volume of dentine decreased from 4.2% to 24.5%.

For maxillary first molars, the volume of dentine decreased from 1.4% to 32.5%, while for second molars, it ranged from 2.5% to 45.3% (Table 3).

### 3.2. Measurment of DT Mandibular Teeth

The loss of dentine tissues of mandibular central incisors varied after root canal preparation between 1.6% and 23%, and for lateral incisors, the volume of excised dentine was from 1.9–23.5%. For canines, the volume of dentinal tissues ranged from 1.2–14%.

For mandibular first premolars, the change in the dentinal thickness of the root was recorded within 6.0–23.2%. For second premolars, measurements revealed that the volume of dentine decreased from 4.2% to 24.5%.

For mandibular first molars, the volume of dentine decreased from 2.9% to 40.3%, while for second molars, it ranged from 2.1–31.8% (Table 4).

### 3.3. Dentinal Thickness in Apical Part

As a result of root canal preparation, in the apical part of the root canal, the minimum amount of dentine tissue was excised in the maxillary lateral incisors along the buccal wall—1.7% in the middle-age group; along the palatal—2.8% in the elderly; along the buccal wall of the palatal root of the maxillary first premolar—2.1% in the elderly group.

For maxillary molars, the minimum amount of dentine tissue was excised in the palatal root of the maxillary first molar along the distal wall; in the young group—1.4% and the middle-age—1.4%, in the palatal root of the second molar along the mesial wall—2.5% in the young group, and in the disto-buccal root of the second molar along the buccal wall—1.7% in the young group.

In the lower jaw, the minimum amount of dentine tissue in the apex area was excised in the central incisors along the buccal wall—1.6% in the young group, in the lateral incisors along the lingual wall—1.9% in the young group, and along the mesial wall—2.2% in middle-aged patients; in canines along the distal wall in the groups of the young—1.2% and elderly—1.3%; in the second premolar along the mesial—0.9% and distal—1.7% walls in the young group; in the distal root of the first molar along the lingual wall, in the groups of middle-age—2.9% and elderly—2.9%, and in the distal root of the second molar along the buccal wall—2.1% in the young group.

As a result of the study, in the coronal third of the root canal, dentine tissue was excised 1.3 times less than in the middle third. The apical third of the root canal was characterized by a decrease in the volume of excised dentine during endodontic treatment by an average of 1.85 times compared with the data in the coronal third and 2.4 times compared with the data in the middle third. The data were presented in Figure 2 and Figure 3.

## 4. Discussion

Dentinal thickness is an important factor to be considered when planning for post-endodontic treatment. The remaining dentine volume could predict the success of endodontic and post-endodontic treatment. This is the first study to evaluate dentinal thickness using CBCT. The dentinal thickness was measured in millimeters from the inner surface of the root canal to the outer surface of the tooth, and the volume of the removed dentine tissues was determined as a percentage.

This study has shown that the coronal and middle thirds of the root canal were the most prepared, which could be related to the endodontic system used to treat the root canal or the presence of a post. Either way, the remaining dentinal thickness was still more than 1 mm in anterior teeth, while in maxillary molar teeth in mesio- and disto-buccal roots and in mandibular molars, the dentinal thickness was less than 1 mm in most cases of intact and endodontically treated teeth, especially in cases of curved roots. This is of great significance and should be considered when preparing these roots for a post after endodontic treatment.

The study showed that the apical part of the root is the least treated, which in turn leads to the development of complications after endodontic treatment and thus explains the high percentage of failures.

The most loss of dentine volume was shown among the maxillary second molar (27%) and mandibular second molar (23.6%), which could be related to the configuration of root canals or the clinical case of these teeth, especially if the procedure was a retreatment.

Gao et al. used micro-CT to measure the minimal remaining dentinal thickness in the mesio-buccal roots of maxillary molars as an attempt to remove broken instruments inside root canals. The authors noticed that the remaining dentine was smaller at the distal walls [24], which coincides with the results of our study, as the largest amount of dentine was removed along the distal walls of mesio-buccal roots, which could be explained by the presence of MB2 in most of the cases or the attempt to extract broken instruments.

In another study, Haralur et al. studied the effect of coronal root canal dentinal thickness on fracture resistance for post treatment and found that a post is needed when DT is less than 1.5 mm [25].

A study by Ha et al. evaluated the remaining dentinal thickness after post preparation and found that post preparation resulted in less than 1 mm of dentine thickness remaining in premolars, smaller roots of molars, and mandibular incisors [[26].

Dentinal thickness is also affected by root canal configuration, especially in molar and premolar areas, which should be considered when choosing the best root for post treatment [27].

## 5. Conclusions

Within the limits of this study, the dentinal thickness in the coronal and middle third of the root decreases significantly more than the apical third. The most loss of dentine volume was among molar teeth, and the remaining dentinal thickness was less than 1 mm, which would increase the risk of complication while preparing the canal for a post.

## Figures and Tables

**Figure 1 dentistry-11-00124-f001:**
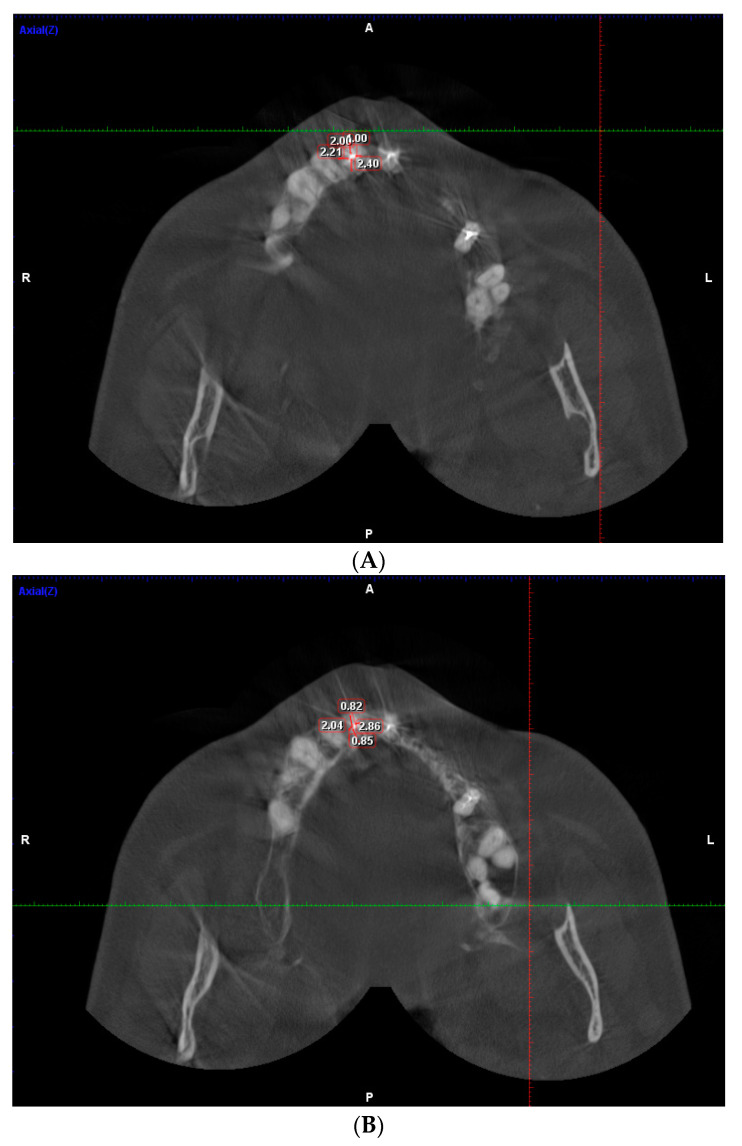
(**A**) Measurement of dentinal thickness around root canal in maxillary central incisor (endodontically treated) coronal third. (**B**) Measurement of dentinal thickness around root canal in maxillary central incisor (endodontically treated) middle third. (**C**) Measurement of dentinal thickness around root canal in maxillary central incisor (endodontically treated) apical third.

**Figure 2 dentistry-11-00124-f002:**
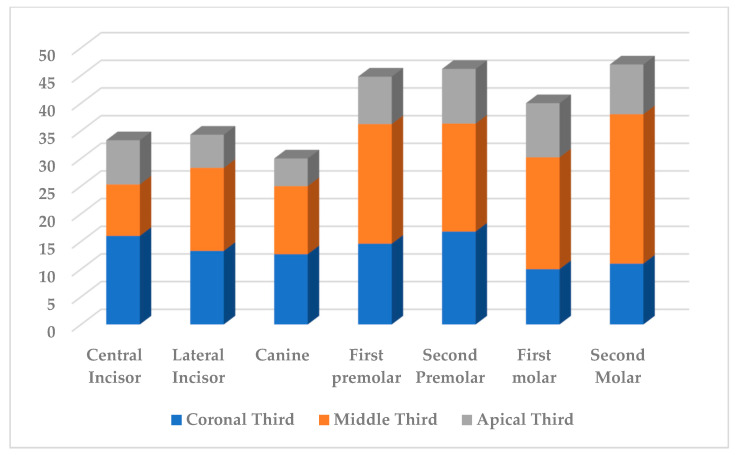
The loss of root canal dentine tissue (%) after mechanical treatment of maxillary teeth.

**Figure 3 dentistry-11-00124-f003:**
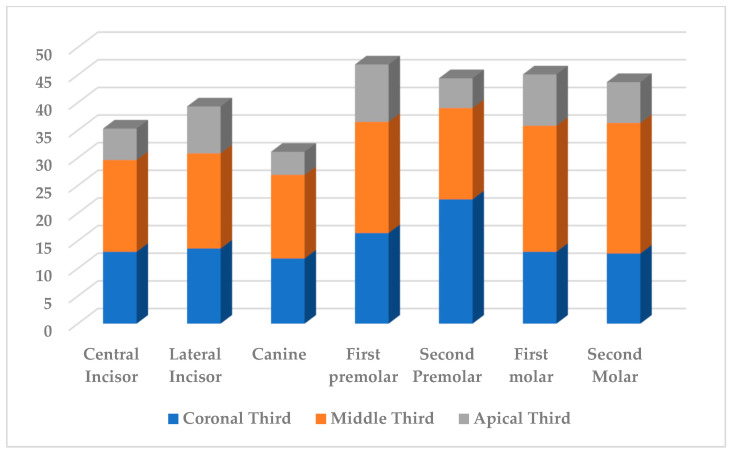
The loss of root canal dentine tissue (%) after mechanical treatment of mandibular teeth.

**Table 1 dentistry-11-00124-t001:** Number of maxillary teeth included in this experiment.

Group	Central Incisors	Lateral Incisors	Canines	First Premolars	Second Premolars	First Molars	Second Molars
Young group	50 intact 48 treated	55 intact 50 treated	55 intact 40 treated	55 intact 48 treated	55 intact 50 treated	50 intact 48 treated	55 intact 43 treated
Middle-age group	55 intact 55 treated	55 intact 55 treated	55 intact 54 treated	55 intact 53 treated	55 intact 55 treated	55 intact 50 treated	50 intact 56 treated
Elderly	55 intact 55 treated	55 intact 54 treated	55 intact 54 treated	55 intact 51 treated	54 intact 55 treated	55 intact 60 treated	47 intact 60 treated

**Table 2 dentistry-11-00124-t002:** Number of mandibular teeth included in this experiment.

Group	Central Incisors	Lateral Incisors	Canines	First Premolars	Second Premolars	First Molars	Second Molars
Young group	55 intact 49 treated	55 intact 48 treated	55 intact 43 treated	55 intact 49 treated	55 intact 48 treated	50 intact 48 treated	55 intact 46 treated
Middle-age group	55 intact 49 treated	55 intact 50 treated	55 intact 51 treated	55 intact 50 treated	55 intact 50 treated	55 intact 53 treated	50 intact 56 treated
Elderly	55 intact 55 treated	55 intact 60 treated	55 intact 60 treated	55 intact 55 treated	50 intact 60 treated	50 intact 60 treated	46 intact 60 treated

**Table 3 dentistry-11-00124-t003:** The remaining dentine volume (%) after root canal treatment for maxillary teeth.

Tooth	Largest Loss of Dentine			Least Amount Loss of Dentine		
	Coronal Third	Middle Third	Apical Third	Coronal Third	Middle Third	Apical Third
Central incisors	Buccal wall 27.7% (Elderly)	Mesial wall 12.1% (Elderly)	Buccal wall 13% (Middle)	Distal wall 9.4% (Young)	Mesial wall 7.8% (young)	Mesial wall 3.6% (young)
Lateral incisors	Palatal wall 14.7% (young)	Palatal wall 20.4% (Middle)	Mesial wall 16.9% (Middle)	Mesial wall 9.5% (young)	Mesial wall 10.3% (middle)	Buccal wall 1.7% (Middle)
Canines	Distal wall 16.7% (Elderly)	Distal wall 17.3% (Middle)	Mesial wall 12.1% (Elderly)	Buccal wall 8.5% (Young)	Buccal wall 11.3% (Elderly)	Distal wall 3.6% (Young)
First Premolar	Distal wall 21.2% (Elderly)	Mesial wall palatal root 31% (Middle)	Buccal wall palatal root 20% (Young)	Palatal wall 12.4% (Elderly)	Distal wall buccal root 14.7% (Middle)	Palatal wall Buccal root 3% (Elderly)
Second Premolar	Mesial wall 22.3% (Elderly)	Mesial wall 24.5% (Middle)	Distal wall 19.2% (Elderly)	Palatal wall 10.7% (Young)	Palatal wall 8.3% (Young)	Buccal wall 4.2% (Young)
First Molar	Distal wall 13.9% (Middle)	Distal wall Mesio-Buccal root 32.5% (Elderly)	Distal wall Mesio-Buccal root 25.5% (Middle)	Mesial wall 6.4% (Middle)	Buccal wall Disto-buccal root 13.5% (Young)	Distal wall Palatal root 1.4% (Young)
Second Molar	Distal wall 14.7% (Elderly)	Mesial wall Mesio-Buccal root 45.3% (Elderly)	Mesial wall Mesio-Buccal root 18.6% (Young)	Palatal wall 8.6% (Young)	Mesial wall Palatal root 13.7% (Young)	Mesial wall Palatal root 2.5% (Young)

**Table 4 dentistry-11-00124-t004:** The remaining dentine volume (%) after root canal treatment for mandibular teeth.

Tooth	Largest Loss of Dentine			Least Amount Loss of Dentine		
	Coronal Third	Middle Third	Apical Third	Coronal Third	Middle Third	Apical Third
Central incisors	Distal wall 19.6% (Elderly)	Distal wall 23% (Middle)	Mesial wall 8.2% (Middle)	Lingual wall 5.4% (Young)	Buccal wall 11.1% (Young)	Buccal wall 1.6% (Young)
Lateral incisors	Mesial wall 19.6% (Elderly)	Distal wall 23.5% (Middle)	Distal wall 15.8% (Elderly)	Lingual wall 7.8% (Young)	Buccal wall 11.8% (Young)	Lingual wall 1.9% (Young)
Canines	Lingual wall 14% (Middle)	Lingual wall 13.4% (Middle)	Lingual wall 7.1% (Middle)	Mesial wall 8.8% (Young)	Mesial wall 9.4% (Young)	Distal wall 1.2% (Young)
First Premolar	Distal wall 21.2% (Elderly)	Lingual wall 23.2% (Elderly)	Mesial wall 19.3% (Elderly)	Mesial wall 14.1% (Middle)	Distal wall 17.7% (Young)	Buccal wall 6% (Elderly)
Second Premolar	Mesial wall 35.4% (Middle)	Mesial wall 21.8% (Young)	Mesial wall 11.8% (Middle)	Buccal wall 12% (Elderly)	Buccal wall 10.1% (Young)	Mesial wall 0.9% (Young)
First Molar	Distal wall 18.3% (Middle)	Mesial wall Distal root 40.3% (Elderly)	Buccal wall Mesial root 15.4% (Elderly)	Mesial wall 9.1% (Middle)	Lingual wall mesial root 16.4% (Young)	Lingual wall Distal root 2.9% (Middle)
Second Molar	Mesial wall 13.6% (Young)	Mesial wall Distal root 31.8% (Elderly)	Mesial wall Distal root 17.2% (Middle)	Buccal wall 11.1% (Young)	Lingual wall Distal root 13.6% (Young)	Buccal wall Distal root 2.1% (Young)

## Data Availability

Not applicable.

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
