# Peer review of "Evaluation of Dentinal Thickness and Remaining Dentine Volume around Root Canals Using Cone-Beam Computed Tomography Scanning"

_dentistry, 2023, doi:10.3390/dj11050124_

Round 1

Reviewer 1 Report

Dear authors

Thank you for your effort. You have done a big amount of work trying to measure all these dimensions. According to my opinion though the results are missed within the extended "results" session. I suggest you to move lines 435-506 of the discussion session into the results session prepare better your tables and delete all you have written in the results. There is too much details that noone could read.  You need two or three new tables where you can put your comparisons and the reader could have an overview of the statistically significant data you are mentioning. You can also use some graphics that could help the reader understand better what you have found. Also in the methodology it is unclear whether one or more members of your team where using the program in order to exclude bias from the use of the system. How where they informed and proceed to calibration etc. Finally the biggest disadvantage of the article presented in this form is the poor introduction and the non existen discussion session. In the discussion you are not explaining why the distal or the palatal part of the root has bigger reduction of the dentine substance for example or what effect has a reduction of more than 20% in the resilience effect of the root etc. You are ony repeating in brief your results. So that is why I suggest you to remove this part to results and rewrite the discussion. Your conclusions are relatively poor and non specific. There is more bibliography to be used and help you in the discussion part.

Author Response

Dear reviewer,

Thank you for your report. We appreciate the effort to review this article, we have modified the article as requested, please kindly review and advise us.

Best regards

Reviewer 2 Report

Dear Authors,

I have read with interest the manuscript as it focuses on a highly relevant topic. It is essential for clinicians to have a solid base for the restoration of endodontic treated teeth and it is well known that fracture is one of the most common complication after root canal treatment often impling invasive procedures such as crown lengthening but in some cases lead to extraction. While the paper addresses an interesting issue, it is not publishable in its current form.

Introduction: The introduction provides a good overview of the topic, but it could benefit from a more clear and concise statement of the research question and objectives. Also it should provide an introduction of the root canal preparations techniques that are now widely used.

Materials and Methods

This section provides a short description of the selection strategy and inclusion criteria. However, it would be helpful to provide more details on the intra-examiner agreement and on the calibration of the operators providing the root canal treatment. Also in my opinion it is mandatory and relevant to include the detailed protocol used in the instrumentation of the endodontic treated teeth. The Methods section does not clearly explain how each tooth was selected for the study and how many teeth were included.

CBCT is associated with a low dose of radiation, still patients should be protected from overexposure to radiations. In my opinion it should be clear the indication for a CBCT pre and post operative. 

Results: The results section provides a reliable summary of the findings, but it could benefit from a more critical analysis of the study. Specifically, it would be helpful to discuss the limitations of the study and the overall quality of the evidence as the method and type of files used in root canal instrumentation may also play a role in dentinal reduction.

Discussion: The discussion section provides a good synthesis of the findings, but it could benefit from a more explicit and focused discussion of the implications of the findings for policy and practice. I would like to see some discussions of the findings of this paper in relation to recent findings and development in endodontology. It should also be considered the type of post to be used post endodontic treatment.

Author Response

(The authors gave the same response as above.)

Round 2

Reviewer 1 Report

Dear authors,

A serious attempt to improve the presentation of your data has been made from your part. After clearing the text from corrections though, you will find certain corrections that need to be made in the "results" part. Also minor text editing needs to be made through out the manuscript.

Thank you for your effort!

Author Response

Dear reviewer,

Thank you for your report. We have revised the result section especially when explaining the dentinal thickness in the apical third.